# Landscape, Socioeconomic, and Meteorological Risk Factors for Canine Leptospirosis in Urban Sydney (2017–2023): A Spatial and Temporal Study

**DOI:** 10.3390/vetsci10120697

**Published:** 2023-12-09

**Authors:** Xiao Lu, Christine Griebsch, Jacqueline M. Norris, Michael P. Ward

**Affiliations:** Sydney School of Veterinary Science, The University of Sydney, Camperdown, NSW 2050, Australia; xiao.lu@sydney.edu.au (X.L.); christine.griebsch@sydney.edu.au (C.G.); jacqui.norris@sydney.edu.au (J.M.N.)

**Keywords:** bacteria, *Leptospira*, dogs, spatial, temporal, mapping, epidemiology

## Abstract

**Simple Summary:**

Leptospirosis is a zoonotic disease that has re-emerged in the Sydney area. We analysed clinical canine leptospirosis cases from 2017 to 2023, from two council areas of urban Sydney, New South Wales, Australia, to identify spatial and temporal risk factors associated with the disease’s transmission. For the spatial risk factors (landscape and socioeconomic factors, seroprevalence, and urban heat island effect), the following three modelling approaches were used: log-transformed Poisson models with an offset of canine population per Level-1 Statistical Area (SA1), conditional logistic regression models adjusted by dog population per SA1, and two families (binomial and Poisson) of a general additive model of smoothed Normalized Difference Vegetation Index (NDVI) and canine leptospirosis. The association with meteorological factors (precipitation and temperature) was tested using multivariate Autoregressive Integrated Moving Average (ARIMA) models. The results indicated that canine leptospirosis is endemic in urban Sydney. Its occurrence was strongly associated with higher community seroprevalence and positively correlated with the presence of a tree-covered areas in the neighbourhood. Clinical cases were more likely to be reported from areas adjacent to veterinary hospitals. More studies should be performed to fully investigate the role of veterinary care services in the occurrence and reporting of leptospirosis, confirm its ubiquitousness in the environment, and identify major wildlife reservoirs in Sydney.

**Abstract:**

Leptospirosis is a potentially fatal zoonotic disease caused by infection with pathogenic *Leptospira* spp. We described reported clinical cases of canine leptospirosis in the council areas of the Inner West and the City of Sydney, Australia, from December 2017 to January 2023 and tested the association with urban spatial (landscape and socioeconomic factors, community seroprevalence, and urban heat island effect) and temporal (precipitation and minimum and maximum temperature) factors and the cases using log-transformed Poisson models, spatially stratified population-adjusted conditional logistic models, General Additive Models (GAMs), and Autoregressive Integrated Moving Average (ARIMA) models. The results suggested that canine leptospirosis is now endemic in the study area. A longer distance to the nearest veterinary hospital (RR 0.118, 95% CI −4.205–−0.065, *p* < 0.05) and a mildly compromised Index of Economic Resources (IER) (RR 0.202, 95% CI −3.124–−0.079, *p* < 0.05) were significant protective factors against leptospirosis. In areas proximal to the clinical cases and seropositive samples, the presence of tree cover was a strong risk factor for higher odds of canine leptospirosis (OR 5.80, 95% CI 1.12–30.11, *p* < 0.05). As the first study exploring risk factors associated with canine leptospirosis in urban Sydney, our findings indicate a potential transmission from urban green spaces and the possibility of higher exposure to *Leptospira*—or increased case detection and reporting—in areas adjacent to veterinary hospitals.

## 1. Introduction

Leptospirosis is a potentially fatal zoonotic disease caused by infection with pathogenic, aerobic, highly motile Gram-negative spirochetes of the *Leptospira* spp. Across all continents, leptospirosis is endemic, and exposure is considered ubiquitous in mammals [1]. Maintenance hosts (e.g., rodents) are usually asymptomatic and can be chronically infected. The bacteria replicate in the host’s proximal renal tubules, resulting in continuous urinary shedding [2]. Once shed by the host, pathogenic leptospires can be very persistent in the environment [3]. By contact of mucous membranes with infectious urine or contaminated soil or water, incidental hosts (e.g., dogs, humans) can be infected, which can cause the acute onset of severe pathologic leptospirosis [1]. The clinical presentation depends on the host’s species and immunity, the virulence of the infecting serovar, and the inoculation dose and, therefore, could vary between individuals even within the same animal species [4]. 

The transmission of pathogenic *Leptospira* to domestic animals and humans could result from spillover from wildlife or livestock reservoirs or a contaminated environment [5]. In a unique, multi-species community, such as in urban areas, transmission could be complicated by local heterogeneity in spatial characteristics and less predictable inter-species interactions [6].

Regarding urban wildlife reservoirs, rats have been the most suspected source of human leptospirosis outbreaks and the most surveyed animals due to their proximity to human households and consistent seropositivity, as well as the microbial presence of pathogenic *Leptospira* in locations including Sydney, Australia [7,8,9,10,11]. In Australia, the presence of *Leptospira interrogans* has also been detected in kangaroos [12], urban possums [13], and flying foxes [14], with evidence of urinary shedding of leptospires from flying foxes [15]. 

Leptospires survive for extended periods of time in water and soil once shed by their hosts [3,16], and, therefore, a lack of an effective drainage system and extreme meteorological events (e.g., heavy rainfall and flooding) could dramatically increase exposure of incidental hosts in an urban context [17,18,19]. Previous human studies closely associated leptospirosis in cities with a lower socioeconomic status, which reflects suboptimal hygiene conditions and an inaccessibility to hospital resources [20,21]. Following a similar logic, it is reasonable to suspect that companion dogs who live in communities with a poorer socioeconomic status could be at higher risk of being infected with *Leptospira*. However, these associations were only described in major cities in developing countries (e.g., Brazil), and few veterinary studies have addressed the risk factors contributing to clinical canine leptospirosis in developed metropolitan areas, where communities of different socioeconomic statuses have similar access to public services [22,23]. 

In December 2017, canine leptospirosis cases re-emerged in Sydney after approximately forty years of an apparent absence of the disease, and the outbreak has been ongoing ever since [24,25]. The Sydney outbreak is characterised by aggressive clinical signs involving acute injury to the liver and kidneys and an overwhelming mortality rate of 89% (2017−2020). A spatiotemporal cluster of clinical cases was detected in suburbs (Surry Hills and Darlinghurst) adjacent to the central business district of the City of Sydney during the winter of 2019 [26]. Patient signalment (age, sex, and breed) did not contribute to the observed clustering. In 2021, a clinical human leptospirosis case in a golf course greenkeeper was reported [27], in which environmental spillover was highly suspected without any causal relationship between human and canine leptospirosis indicated. Although it was hypothesised that the strains underlying the current Sydney outbreak are more virulent [24], spillover from wildlife reservoirs and the environmental pool of pathogenic *Leptospira* spp. still likely played important roles in transmission. Moreover, knowledge of the effects of the urban landscape and socioeconomic factors on clinical canine leptospirosis in a metropolitan environment is absent, and previous studies conducted in other geographical locations should not be directly applied to Sydney due to its unique urban morphology, social structure, veterinary practice, and climate. It is necessary to scan both spatial and temporal risk factors of interest in an exploratory study and then investigate other unidentified exposure risks for pathogenic *Leptospira* spp. in this specific geographical area. 

In the current study, we aimed to achieve the following: 1. describe the spatial and temporal pattern of clinical canine leptospirosis cases in Sydney from December 2017 to January 2023; and 2. assess the association between landscape factors (land use, presence of tree coverage, Normalized Difference Vegetation Index, distance to flying fox colony), socioeconomic factors (distance to veterinary hospital, socioeconomic indices), local meteorological factors (urban heat island effect, weekly rainfall, and weekly average maximum and minimum temperature), seroprevalence in unvaccinated dogs and the reported clinical canine leptospirosis cases.

## 2. Materials and Methods

### 2.1. Data Collection and Cleaning

De-identified canine leptospirosis data in Sydney, New South Wales (NSW), from December 2017 to January 2023 were acquired [24]. The cases were presented to various veterinary hospitals for acute onset of marked clinical presentations (predominantly featuring vomiting, lethargy, icterus, and abdominal pain, while serum biochemistry indicated profound hepatorenal damage). The infection was confirmed subsequentially using either the polymerase chain reaction (PCR) method on urine and/or blood and/or a microscopic agglutination test (MAT) titre ≥1:800 in a non-vaccinated dog and reported directly by the primary veterinarians. The residential address of each case was geocoded manually to coordinates in decimal degrees using Google Earth [28]. Spatial cropping, mapping, and spatial analytic processes were performed in ArcGIS Pro 2.5.0 [29], SaTScan v 10.1.2 [30], and Rstudio interface 2023.06.1+524 (R version 4.2.2) [31]. 

The study area comprised the local government areas of the City of Sydney and the Inner West Council. The administrative boundaries and their subordinate Statistical Areas Level 1 (SA1s) were accessed from the Australian Statistical Geography Standard (ASGS) Edition 3 (July 2021–June 2026) [32]. The population of human residents per SA1 was downloaded as part of the 2021 census and used to exclude non-residential SA1s from the study [33]. The centroid coordinate of each residential SA1 was used as a location for spatial analytics. Considering the increase in new dog ownership during the COVID pandemic, the 2021 dog population in the City of Sydney was preliminarily estimated as 1.2 times the count of registered dogs per SA1 in the City of Sydney in 2020 [25,34]. An appropriate ratio between the human and dog population per SA1 in the City of Sydney was calculated and then applied to residential SA1s in the Inner West Council. Finally, empirical Bayesian kriging (smooth factor 0.5) was performed on the estimates to create the final estimation of canine population per SA1 in the study area. 

Two main categories of spatial risk factors (landscape and socioeconomic) were included in the study, and canine seroprevalence of pathogenic *Leptospira* spp. in the neighbourhood was also included as an extra variable in the statistical tests on the landscape and socioeconomic factors. The landscape factors included the distance to potential wildlife reservoirs (e.g., the nearest flying fox colonies in urban Sydney), the distance to the nearest off-leash dog park, the presence of tree cover in the SA1, the presence of a commercial facility in the SA1, and the presence of a recreational and cultural facility in the SA1. 

The geographic coordinates of the flying fox colonies in the City of Sydney were obtained from the National Flying-fox Monitoring Program [35]. The coordinates of the off-leash dog parks in the study area were adapted from the dataset and list published by the two councils [36,37]. The areas of tree cover within SA1 in 2018 were first calculated from the NSW native vegetation raster dataset (version 1.20) [38] using the Calculate Geometry function in ArcGIS Pro and then converted to a dichotomous variable (presence or absence) based on the results. Concurrently, a raster of Normalized Difference Vegetation Index (NDVI) of the study area was generated from the Landsat 8 Collection 2 Tier 1 satellite image set [39], on the Google Earth Engine. The NSW land use raster (version 1.20) published in 2020 was used to manually identify commercial and recreational and cultural facilities in each SA1 [40]. 

In terms of socioeconomic factors, the distance to the nearest veterinary hospital, the Index of Economic Resources (IER) of the SA1, and the Index of Education and Occupation (IEO) of the SA1 were included. The Veterinary Practitioners Board of New South Wales website was accessed to retrieve the addresses of veterinary hospitals, which were then manually geocoded [41]. The IER and IEO scores were also acquired as part of the 2021 census [42]. 

A higher retrospective seroprevalence of pathogenic *Leptospira* spp. in the canine population was estimated by generating 1km buffers around the positive, unvaccinated canine samples of various signalments and breeds (low positive MAT titre ≥ 1:50 against at least one specific serovar) from a sero-survey performed in 2019 [43]. The survey was completely independent from the current study, and we accessed its retrospective, de-identified data to represent seroprevalence as a potential risk factor. Only samples with a locatable residential address were mapped to create the buffers. Being within one of the buffers or not was defined as a dichotomous variable to present a risk of higher environmental exposure in the residential SA1. 

While all the residential SA1s were included in the following analyses, a subset of SA1s covered by a 1km diameter of geodesic buffers surrounding the SA1s with at least one clinical canine case was generated to minimise variability in exposure to pathogenic leptospires. From these buffers, control SA1s were randomly selected and frequency-matched to case SA1s (four controls to one case in each buffer). 

Lastly, the Urban Heat Island Index (UHII) per mesh block in 2016 was accessed as a purely spatial factor presenting the urban microclimate [44]. 

All the variables were spatially joined to the SA1s and then exported as .csv files into Microsoft Excel [45]. One IER value of zero was excluded from subsequent tests of the variable. A conversion from the numeric variables (distances, IER, and IEO scores) to the categorical variables was performed due to the non-linear distribution of the data. 

For the temporal analysis, the daily precipitation (mm) measured at the Botanic Gardens station during the study period was retrieved from the Bureau of Meteorology database [46] and used to calculate the weekly precipitation (mm) in the study area. The daily maximum and minimum temperature measured at the Observatory Hill station during the study period was accessed from the same database, from which the weekly average maximum and minimum temperatures were calculated. 

### 2.2. Spatial Statistical Analyses

Poisson models of the spatial scan statistic and monthly space–time scan statistic scanning for locations with high rates (23 December 2017 to 13 January 2023) were conducted using 999 sets of Monte Carlo stimulation in SaTScan to identify statistically significant (*p* < 0.05) spatial and spatiotemporal clustering.

The cleaned .csv datasheets were read and analysed in the RStudio interface. All the statistical models were built with stepwise algorithms, and their Akaike Information Criterion (AIC) was calculated to compare the models and identify the best set of predictors. 

For the Poisson regression, all the residential SA1s were included in the analysis and the glm() function from the stats Rpackage was used for modelling. The log-transformed SA1 dog population was added as an off-set to every Poisson model produced. The count of canine cases per SA1 was used for incidence rate calculations instead of a dichotomous variable. First, univariate Poisson models were created and variables with a *p*-value less than or equal to 0.20 were selected for multivariate models. Two multivariate Poisson models (landscape factors and socioeconomic factors) were developed using forwards stepwise algorithms; seroprevalence was added as a predictor if significance was shown by the *p*-value and/or lower AIC score of the model. 

For the conditional logistic regression, the selected case- and frequency-matched control SA1s were included and stratified using individual 1km buffers. The clogit() function from the survival R package was used for modelling. Seroprevalence was not included in the conditional logistic models because the buffers created around the seropositive samples highly overlapped with the case–control buffers. All the models were adjusted by dog population per SA1, and the variables with a *p*-value less or equal to 0.25 from the univariate models were made available for the multivariate models. Two multivariate conditional logistic regression models (landscape factors and socioeconomic factors) were developed using a forward stepwise algorithm. 

Additionally, an univariable log-transformed Poisson model with SA1 canine population offset and a buffer-stratified canine-population-adjusted conditional logistic model was developed for UHII per SA1 and clinical cases using the protocol described above. Based on the NDVI raster and on the case per SA1, two spatial Generalized Additive Models (GAMs) from the binomial and Poisson family were developed, respectively, using the gam R package.

The autocorrelation of residuals was tested for each final model using the acf() function in the stats R package.

### 2.3. Temporal Statistical Analyses

Initially, weekly and monthly time series of clinical canine leptospirosis cases during the study period were generated using Rstudio. The time series were then decomposed and visualised to identify any trend or seasonality. The stationarity of the time series was also confirmed using plots, the Ljung–Box test, the Augmented Dickey–Fuller (ADF) test, and the Kwiatkowski–Phillips–Schmidt–Shin (KPSS) test. By using the auto.arima() function from the forecast package and manual adjusting the terms with the assistance of the Autocorrelation Function (ACF) and Partial Autocorrelation Function (PACF) plots of both time series, Autoregressive Integrated Moving Average (ARIMA) models were developed to assess the autocorrelation present in the time series [47]. The AIC scores were used to compare the quality of all the models.

The time series of weekly precipitation and average minimum and maximum temperature were tested, separately, for stationarity prior to being included as covariates when developing the multivariate ARIMA models. A specific length of time series (from the fourth to the seventieth week) was selected as a training dataset to determine the best weekly lag between the cases and the climate time series using the AIC scores. The lags were then applied back to the multivariate ARIMA models developed using auto.arima().

## 3. Results

From December 2017 to January 2023, 20 canine leptospirosis cases were confirmed in 18 SA1s in council areas of the City of Sydney and the Inner West (Figure 1). The clinical cases were spatially clustered in the suburbs of Surry Hills and Redfern in the City of Sydney (covered 29 SA1s, RR 19.21, *p* < 0.001). North of the spatial cluster, a spatiotemporal cluster of high rates (six cases) was detected in Surry Hills and Darlinghurst during the autumn, winter, and spring of 2019 (covered 56 SA1s, RR 66.62, *p* < 0.001). 

Temporally, the case time series showed no seasonality (four cases observed in the spring, five in the summer, three in the autumn, and eight in the winter) nor trend. Stationarity was indicated in both time series by non-significant Ljung–Box test and KPSS test (*p* > 0.05), despite the ADF test for both time series failing to reject the null hypothesis of non-stationarity (*p* > 0.5). The weekly ARIMA (0,0,1) model without seasonality was the best fit (AIC 79.61) of the case time series and autocorrelation was not detected in the monthly time series. The clinical canine leptospirosis cases were autocorrelated with cases at 5, 20, and 26 weeks (approximately in one month and in five or six months) (Figure 2). 

From the included residential SA1s (*n* = 906), 72 control SA1s were randomly selected from 1km buffers around the 18 case SA1s for the conditional logistic regression analysis (Figure 3). There were 28 veterinary hospitals in the study area. Tree cover across the study area and the final converted dichotomous values are illustrated in Figure 4a and the maximum NDVI per SA1 in Figure 4b. Thirteen seropositive samples were included in this study, out of which four samples tested MAT-positive for serovar Australis (one sample co-infected with serovar Cynopteri), three samples positive for serovar Copenhageni (one sample co-infected with Javanica), two samples positive for serovar Djasiman, and four samples positive for serovar Topaz. 

Autocorrelation of residuals was not present in any of the final models. 

### 3.1. Log-Transformed Poisson Regression Models

As indicated by the univariate (Table 1, Table 2 and Table 3) and multivariate Poisson regression models (Table 4 and Table 5), seroprevalence was an important risk factor and was included in both final models. In the final landscape model, seroprevalence was also highly statistically significant (RR 2.955, 95% CI 0.127–2.040, *p* < 0.05). 

In the final socioeconomic model, a relatively long distance (>851 m) to the nearest veterinary hospital (RR 0.118, 95% CI −4.205–0.065, *p* < 0.05) and a mildly compromised IER (>884, ≤955) of the SA1 (RR 0.202, 95% CI −3.124–−0.079, *p* < 0.05) contributed significant protection against the risk of canine leptospirosis. 

No statistical association was found between the UHII per SA1 and the incidence rates in the Poisson model (*p* > 0.05). 

### 3.2. AIC Comparison—Landscape and Socioeconomic Factors

For the final Poisson models and the conditional logistical models, lower AIC scores were noted in the socioeconomic models compared to their landscape counterparts. A model residual map of the final multivariate socioeconomic Poisson model is shown in Figure 5.

### 3.3. Conditional Logistic Models Adjusted by Canine Population

Table 6 and Table 7 show the univariate models, and Table 8 and Table 9 summarise the final multivariate landscape and socioeconomic models. 

In the areas proximal to the case SA1s, the presence of tree cover was a strong risk factor for canine leptospirosis (OR 5.797 95% CI 1.116–30.106, *p* < 0.05). In terms of the socioeconomic factors, the IER (>876, ≤931) was a significant protective factor (OR 0.024, 95% CI 0.001–0.585, *p* < 0.05). 

No statistical association was found between the UHII per SA1 and canine leptospirosis in the conditional logistic model (*p* > 0.1). 

### 3.4. GAM of NDVI

Both families of GAM showed no parametric nor non-parametric association between the smooth term (NDVI) and the canine leptospirosis cases (parametric binomial *p* = 0.877, non-parametric binomial *p* = 0.641; parametric Poisson *p* = 0.655, non-parametric Poisson *p* = 0.653). 

### 3.5. Multivariate ARIMA Models

None of the meteorological covariates (weekly precipitation, differenced weekly average minimum and maximum temperature) contributed to the autocorrelation of the case time series (ARIMA (0,0,0) showed the best fit to all three covariates, indicated by the lowest AIC score compared to the other ARIMA models: an AIC of 68.71 for the precipitation with one lag term, an AIC of 56.15 for the weekly average minimum temperature with one lag and one difference term, and an AIC of 60.13 for the weekly average maximum temperature with one lag and one difference term).

## 4. Discussion

This is the first study to investigate the association between spatial and temporal risk factors and reported canine leptospirosis cases since its re-emergence in urban Sydney. We found that canine leptospirosis was strongly related to a higher seroprevalence in some neighbourhoods, which could indicate a higher exposure to *Leptospira* spp. resulting in clinical leptospirosis. Within these suspected high-risk areas, a higher coverage of trees increased the risk of leptospirosis. In general, socioeconomic status, including the distance to veterinary care and to economic resources, was more closely associated with the occurrence of cases compared to landscape features (environmental exposure associated with flying fox camps, environmental exposure from off-leash dog parks, and urban land use). The significant protective effects of a longer distance to the nearest veterinary hospital on canine leptospirosis cases also suggests either a contamination of the veterinary clinical environment or possible underreporting in the two council areas, as well as in places further away from Sydney’s central business district. In terms of the temporal aspect of the clinical cases, the 56 SA1s of Surry Hill and Darlinghurst remained the only significant spatiotemporal cluster in 2019 [26]. Autocorrelation in the short term (5, 20 or 26 weeks) was observed in the time series of the clinical cases, but neither precipitation nor temperature contributed to it. The occurrence of cases was non-seasonal and relatively random, and, thus, we conclude that canine leptospirosis has become endemic in urban Sydney. 

In the current study, we analysed canine leptospirosis cases diagnosed and reported by practicing veterinarians from areas governed by the City of Sydney Council and the Inner West Council, which are the most central locations of this metropolis. Clinical canine leptospirosis has been observed from more suburban areas across the Greater Sydney region, and the current selection of the study area was made because a known spatiotemporal cluster existed within it [26]. No further spatial breakdown was performed as our study area of urban Sydney is already of a fine geographical scale. The cases themselves were either significantly clustered or closely adjacent to a significant cluster, and, thus, separated from other more randomly and dispersed cases distributed in Greater Sydney; this minimised spatial heterogeneity and potential confounding. To take neighbourhood-level variations in exposure into account, we stratified the neighbourhoods spatially by creating eighteen 1km buffers and implementing conditional logistic regression models. However, our assumption that dog owners and their dogs access facilities (e.g., off-leash parks, veterinary hospitals) in proximity more frequently was a prerequisite of some of the hypothesized risk factors analysed in this study. This assumption might not be entirely true in real-world scenarios as people’s preferences are influenced by other complex, unmeasurable factors, e.g., community crime rates [48], but the assumption should not be rejected immediately due to the lack of related studies in southeastern Australian cities. It was notable that the majority of residential SA1s had convenient access to both off-leash dog parks (the upper quartile of distance to the nearest park was 550 m) and veterinary hospitals (the upper quartile of distance to the nearest veterinary hospital was 851 m). We do not eliminate the possibility that dogs travel longer distances to access facilities with greater amenities and thar the destination choice is associated with human demographics and behaviours at the individual level [49]. While inclusion of socioeconomic indices in the models gave some insight into the effect of these unmeasured factors, we caution that the current findings are exploratory and owner behaviours should be surveyed and analysed.

As shown in Figure 1 and Figure 3, the two council areas are heavily resided, and urban green space was diffusely present within the residential blocks. As a numeric measurement of vegetation vigour, the NDVI was introduced to this study to compensate for the misclassification error of the dichotomous variable “presence of tree cover in the SA1” caused by spatial analytics. In a study performed in Kansas and Nebraska, USA, living in proximity (≤2500 m) to public parks was a significant risk factor for canine leptospirosis [50]. Although our study was conducted in noncomparable geographical settings and we are yet to conclusively identify the role of off-leash dog parks in leptospirosis transmission in Sydney, we partially agree with the USA study mentioned above in that residing adjacent to forests (an area covered by trees) could increase the risk of canine leptospirosis. Overall, both approaches indicated that the occurrence of cases was not associated with tree cover on a large geographical scale, but the disease is likely transmitted in a community’s green space or is associated with certain activities in these discrete, vegetated areas in communities with reported cases and higher unvaccinated seroprevalence. In conjunction with the absence of an association between the climate events (precipitation and temperature) and cases throughout the study period, as well as the minimum flooding risk in the two council areas [51], transmission may be more dependent on fixed stagnant water resources (e.g., public water bowls for dogs) [52,53] and ubiquitous urban wildlife activity compared to environmental spillover alone. This study’s findings indicate the necessity for environmental and wildlife sampling in community green spaces to quantify exposure, as *Leptospira* spp. can survive in soil and water for up to six months [52]. 

Regarding exposure to wildlife reservoirs, we investigated flying foxes as a potential source of pathogenic *Leptospira*, and no association between the exposure to flying foxes and the cases was found. There are three flying fox colonies within or near the study area. In one location (Botanic Gardens), flying foxes have not been observed for years, whereas both grey-headed flying foxes (*Pteropus poliocephalus*) and black flying foxes (*Pteropus alecto*) have been consistently observed in the other two locations (Centennial Park and Girrahween Park) [35], which are both off-leash dog parks where dogs can be in direct contact with flying foxes or exposed to an environment contaminated by flying foxes’ urine. Urban flying foxes travel 2–30 km to forage [54,55], and they more commonly graze in areas with denser tree cover [55]. They may cause a more significant impact in locations outside our study area. Previously, pathogenic *Leptospira* has been identified using PCR in 8% of sampled rats [8]. Due to the lack of knowledge about rat populations in Sydney, the presence of commercial properties (e.g., restaurants) in the SA1s was included to present neighbouring rat presence and activity [56]. There was no association between commercial properties and canine leptospirosis, but it is certain that prospective surveys on rats in the urban Sydney are of high priority.

With respect to the socioeconomic factors, the reported cases were only negatively associated with mildly compromised IER scores (from lower quartile to median), and no difference was detected between the SA1s with severe poverty (less than or equal to the lower quartile) and the wealthier SA1s (more than the median). Canine leptospirosis was also not associated with the owners’ educational and occupational backgrounds. Several studies have demonstrated that a higher education level significantly encouraged a positive attitude and better preventative practices towards leptospirosis as a zoonotic disease in humans [57,58]. Therefore, it may be logical to assume that a similar association also exists between an owner’s education level and attitude towards veterinary prevention against canine leptospirosis. One hypothesis explaining the absence of this association in urban Sydney would be that the overall IEO status in the study area was consistently advantaged and lacked variation, which might otherwise unmask the potential association between the factors and clinical canine leptospirosis. Most SA1s in the City of Sydney and the Inner West had IEO scores higher than the 80 percentile of the national IEO per SA1 in 2021 [42], which suggests that the owners in the study area had more knowledge regarding zoonotic disease and that they were more keen on pursuing veterinary prevention against canine leptospirosis. The scenario may be very different in suburban areas of Western Sydney (e.g., Bardia, where one clinical case was reported in April 2023 [37]) or in rural areas in New South Wales, where the population is at an educational and occupational disadvantage. 

In contrast to the overall advantaged educational level, poverty in the two council areas was prevalent, as the residents in many SA1s received lower incomes and their IER scores were less than the lower quartile of the national SA1 IER scores [42]. A strong association between poverty (low IER scores) and canine leptospirosis was expected because of presumed poor sanitation and consequential exposure to a contaminated environment in human populations [20,59]. The high incidence of canine leptospirosis in SA1s with higher IER scores may be explained by more tree canopy and, therefore, more habitable spaces for wildlife reservoirs in wealthier communities [60]. However, in many cities, community tree cover is not essentially associated with a higher socioeconomic status but is more closely related to historic urban morphology and residents’ lifestyle [61,62,63]. In summary, exposure to *Lepstospira* in dogs appears to be more of an event that happens at the individual level. Community-level economic parameters were not an ideal predictor in terms of assessing the risk factors contributing to the re-emergence of canine leptospirosis in Sydney; further breakdown of an animal and its owner’s lifestyle and residential environment are required.

The negative correlation between the distance to the nearest veterinary hospital (>851 m) and the reported cases raised the following two concerns: a higher exposure to *Lepstospira* around hospitals, and unknown diagnostic and reporting behaviours of primary-care veterinarians. As a rare, re-emerging, and non-notifiable disease, it is not unrealistic for local primary-care veterinarians to neglect the risk of leptospirosis and fail to perform proper biosecurity measures to protect themselves, other veterinary staff, clients, and animals against such a negligible risk. A recent survey conducted in Arizona, USA, identified a lack of leptospirosis-related clinical knowledge in their local veterinary professionals [64]; a similar gap in clinician education might be present in Sydney as well. Furthermore, considering the acute onset and high mortality observed in the local cases [24] and the low IER scores in the Inner West and in the City of Sydney, we suspect the possibility of a failure of diagnosis and a consequential failure of voluntary notification to colleagues and the public, due to external factors including a client’s financial constrictions. There is a need to consider that *Leptospira* spp. may be present around veterinary hospitals, and canine leptospirosis should be considered endemic in Sydney.

In the current study, we were only able to access dog registration data from the City of Sydney, and their counterpart in the Inner West council area were not available upon request. We had to estimate the canine population using incomplete data that might have been underestimated, as only 55.6% of the underlying microchipped population was registered in the City of Sydney [26], and this proportion can vary across council areas. Secondly, the geographical scale applied in the current study caused difficulty in retrieving more accurate forms of data. The exploratory nature of the current study and the use of secondary data has limited our investigation of highly suspected sources of *Leptospira* (stagnant waterbodies and rats). Comprehensive field work is needed for the next step of breaking down risk factors associated with canine leptospirosis, and this cannot be conducted successfully without more public attention on zoonotic disease. Finally, the data used in the models did not represent dynamic changes in urban Sydney, and we assumed that minimal changes happened to the urban morphology of Sydney between 2017 and 2023. 

Nevertheless, our study identified the potential ubiquitous exposure of pathogenic *Leptospira* in urban Sydney regardless of seasonality and rainfall and that transmission occurs in urban green spaces. We cannot find sound reasoning behind the protective effect of distance to veterinary hospitals to clinical canine leptospirosis at this stage, but we call for veterinarians’ awareness of potential exposure to *Leptospira* around their clinics and acknowledge that there might have been an underestimation of the true incidence of clinical cases. 

## 5. Conclusions

From 2017 to 2023, clinical canine leptospirosis cases have been spatially clustered in the City of Sydney. The cases were endemic in nature, with no seasonality observed. The occurrence of clinical cases was significantly associated with positive canine serosamples from healthy, unvaccinated dogs. Within the areas with a higher seroprevalence, the transmission of *Leptospira* likely happened in community green spaces. A higher exposure to *Leptsopira* in and near veterinary hospitals or a higher reporting rate of cases near veterinary hospitals is suspected. There is a need for environmental and wildlife surveys as well as investigations of owners’ lifestyle and veterinarians’ perception towards canine leptospirosis in urban Sydney.

## Figures and Tables

**Figure 1 vetsci-10-00697-f001:**
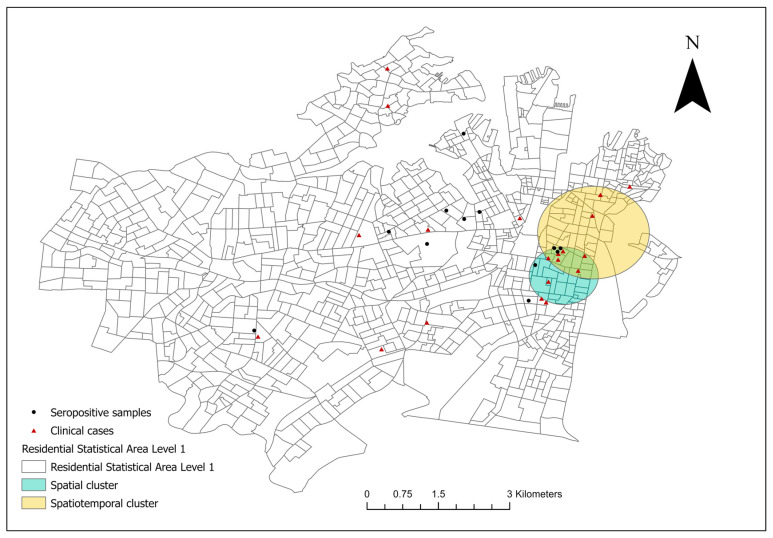
The spatial distribution of the residential addresses of reported clinical canine leptospirosis cases [*n* = 20, distributed in 18 Level-1 Statistical Areas (SA1s)] in council areas of the City of Sydney and the Inner West from December 2017 to January 2023. A significant, purely spatial cluster of clinical cases including 29 SA1s was identified using spatial scan statistic (teal circle); a significant spatiotemporal cluster was present from the 1st of May to the 30th of November 2019 (yellow circle). The seropositive samples were healthy, unvaccinated canine serum samples (MAT titre ≥ 1:50) collected in 2019.

**Figure 2 vetsci-10-00697-f002:**
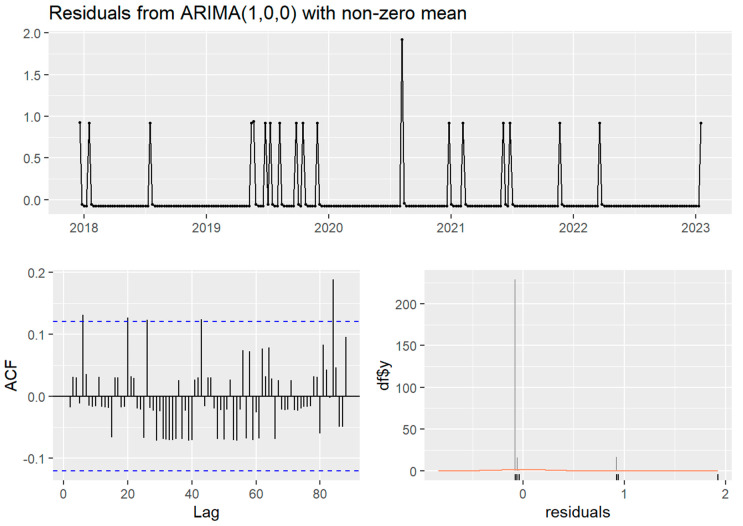
The Autocorrelation Function (ACF) and residual plots of ARIMA (1,0,0) without seasonality of the weekly time series of clinical canine leptospirosis cases (*n* = 20) from December 2017 to January 2023. The blue dashed line indicates statistical significance. The orange line demonstrated normal distribution.

**Figure 3 vetsci-10-00697-f003:**
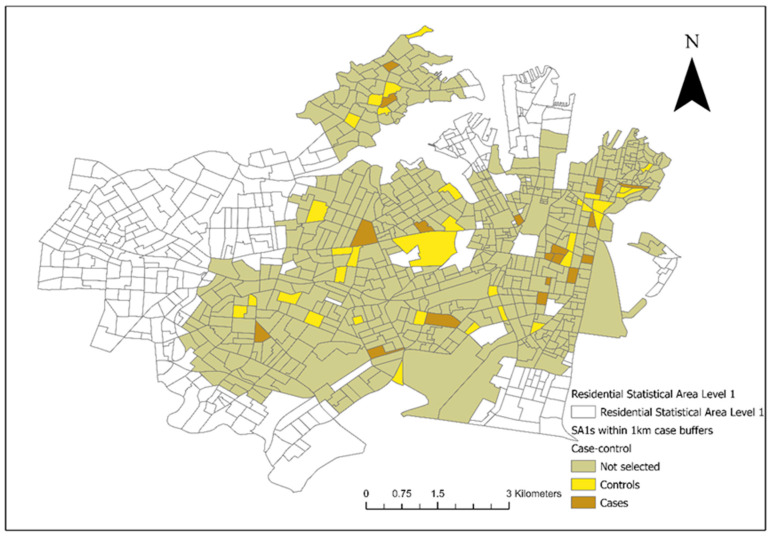
Selected case and control Statistical Areas (Level 1) (SA1s) (18 cases and 72 controls) from areas within a 1km diameter from the SA1s where the clinical canine leptospirosis patients resided in council areas of the City of Sydney and the Inner West from December 2017 to January 2023.

**Figure 4 vetsci-10-00697-f004:**
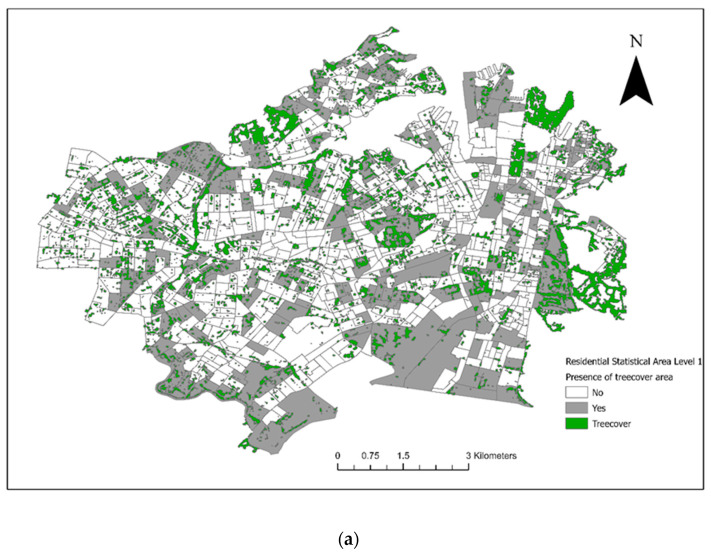
(**a**) Tree cover within council areas of the City of Sydney and the Inner West in 2018. (**b**) The maximum Normalized Difference Vegetation Index (NDVI) per Statistical Area (Level 1).

**Figure 5 vetsci-10-00697-f005:**
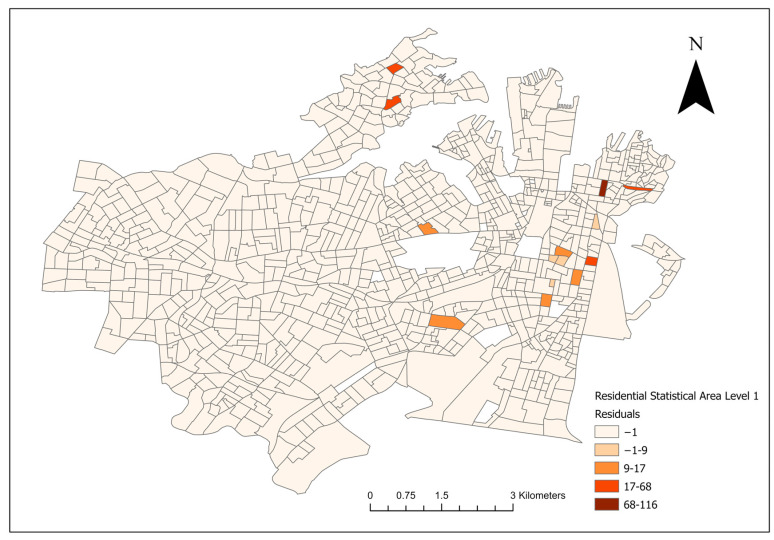
Residual map of the final multivariate socioeconomic log-transformed Poisson model of canine leptospirosis from December 2017 to January 2023.

**Table 1 vetsci-10-00697-t001:** Univariate log-transformed Poisson models of canine leptospirosis and landscape factors from all the residential Statistical Areas (Level 1) in the areas of the Inner West and the City of Sydney (*n* = 906) from December 2017 to January 2023.

Variables		Count of SA1s	Risk Ratio (95% CI)	SE	Z Value	*p* Value
Distance to the nearest flying fox colony (m)	≤2252	4 (223)				
	>2252, ≤3272	7 (219)	2.014 (−0.478–−1.878)	0.601	1.165	0.244
	>3272, ≤4500	5 (221)	1.128 (−1.194–1.436)	0.671	0.180	0.857
	>4500	2 (225)	0.473 (−2.447–0.948)	0.866	−0.865	0.387
Distance to the nearest off-leash dog park (m)	≤233	5 (222)				
	>233, ≤354	5 (221)	0.864 (−1.333–1.041)	0.606	−0.241	0.810
	>354, ≤550	6 (220)	1.208 (−0.901–1.279)	0.556	0.340	0.734
	>550	2 (225)	0.362 (−2.616–0.585)	0.817	−1.244	0.214
Area of tree cover in the SA1	No	12 (698)				
	Yes	6 (190)	1.873 (−0.291–1.546)	0.469	1.338	0.181
Land for a commercial facility in the SA1	No	8 (371)				
	Yes	10 (517)	0.879 (−1.010–0.751)	0.450	−0.288	0.773
Land for a recreational and cultural facility in the SA1	No	12 (496)				
	Yes	6 (392)	0.670 (−1.319–0.519)	0.469	−0.854	0.393

**Table 2 vetsci-10-00697-t002:** Univariate log-transformed Poisson models of canine leptospirosis and socioeconomic factors from all the residential Statistical Areas (Level 1) in the areas of the Inner West and the City of Sydney (*n* = 906) from December 2017 to January 2023.

Variables		Count of SA1s	Risk Ratio (95% CI)	SE	Z Value	*p* Value
Distance to the nearest veterinary hospital (m)	≤373	8 (219)				
	>373, ≤587	7 (219)	0.893 (−1.065–0.840)	0.486	−0.232	0.816
	>587, ≤851	2 (224)	0.243 (−2.947–0.117)	0.782	−1.810	0.070
	>851	1 (226)	0.130 (−4.103–0.029)	1.054	−1.932	0.053
The Index of Economic Resources (IER)	≤884	8 (220)				
	>884, ≤955	2 (222)	0.192 (−3.170–−0.134)	0.775	−2.139	0.032 ^1^
	>955, ≤1008	5 (224)	0.472 (−1.825–0.322)	0.548	−1.381	0.167
	>1008	3 (221)	0.292 (−2.520–0.060)	0.658	−1.877	0.061
The Index of Education and Occupation (IEO)	≤1123	5 (222)				
	>1123, ≤1161	7 (230)	0.926 (−1.124–0.971)	0.535	−0.144	0.886
	>1161, ≤1187	3 (217)	0.413 (−2.237–0.467)	0.690	-1.283	0.200
	>1187	3 (219)	0.408 (−2.250–0.455)	0.690	−1.301	0.193

^1^ *p*-value is less than 0.05.

**Table 3 vetsci-10-00697-t003:** Univariate log-transformed Poisson models of canine leptospirosis and canine seroprevalence from all the residential Statistical Areas (Level 1) in the areas of the Inner West and the City of Sydney (*n* = 906) from December 2017 to January 2023.

Variables		Count of SA1s	Risk Ratio (95% CI)	SE	Z Value	*p* Value
Distance to a canine positive serological sample (m)	>500	6 (503)				
	≤500	12 (385)	2.908 (0.111–2.024)	0.488	2.188	0.029 ^1^

^1^ *p*-value is less than 0.05.

**Table 4 vetsci-10-00697-t004:** Multivariate log-transformed Poisson models of canine leptospirosis, landscape factor, and seroprevalence from all the residential Statistical Areas (Level 1) in the areas of the Inner West and the City of Sydney (*n* = 906) from December 2017 to January 2023. The Akaike Information Criterion (AIC) of the model was 191.647.

Variables		Count of SA1s	Risk Ratio (95% CI)	SE	Z Value	*p* Value
Distance to a canine positive serological sample (m)	>500	6 (503)				
	≤500	12 (385)	2.955 (0.127–2.040)	0.488	2.220	0.026 ^1^
Area of tree cover in the SA1	No	12 (698)				
	Yes	6 (190)	1.933 (−0.260–1.578)	0.469	1.405	0.160

^1^ *p*-value is less than 0.05.

**Table 5 vetsci-10-00697-t005:** Multivariate log-transformed Poisson models of canine leptospirosis, socioeconomic factors, and seroprevalence from all the residential Statistical Areas (Level 1) in the areas of the Inner West and the City of Sydney (*n* = 906) from December 2017 to January 2023. The Akaike Information Criterion (AIC) of the model was 187.635.

Variables		Count of SA1s	Risk Ratio (95% CI)	SE	Z Value	*p* Value
Distance to a canine positive serological sample (m)	>500	6 (503)				
	≤500	12 (385)	2.492 (−0.090–1.916)	0.581	1.371	0.075
Distance to the nearest veterinary hospital (m)	≤373	8 (219)				
	>373, ≤587	7 (219)	0.884 (−1.076–0.830)	0.486	−0.253	0.800
	>587, ≤851	2 (224)	0.236 (−2.979–0.093)	0.784	−1.841	0.066
	>851	1 (226)	0.118 (−4.205–−0.065)	1.056	−2.022	0.043 ^1^
The Index of Economic Resources (IER)	≤884	8 (220)				
	>884, ≤955	2 (222)	0.202 (−3.124–−0.079)	0.777	−2.061	0.039 ^1^
	>955, ≤1008	5 (224)	0.481 (−2.979–0.093)	0.553	−1.324	0.186
	>1008	3 (221)	0.413 (−2.239–0.469)	0.691	−1.281	0.200

^1^ *p*-value is less than 0.05.

**Table 6 vetsci-10-00697-t006:** Univariate conditional logistic models of canine leptospirosis and landscape variables, adjusted by dog population per Statistical Areas (Level 1) (SA1) in 1km buffers around case SA1s (*n* = 18) from December 2017 to January 2023.

Variables		Count of SA1s	Odds Ratio (95% CI)	SE	Z Value	*p* Value
Distance to the nearest flying fox colony (m)	≤2780	10 (34)				
	>2780	8 (38)	0.735 (0.049–10.995)	1.380	−0.223	0.824
Distance to the nearest off-leash dog park (m)	≤199	5 (16)				
	>199, ≤294	3 (19)	0.790 (0.153–4.078)	0.837	−0.281	0.779
	>294, ≤376	3 (20)	0.949 (0.167–5.379)	0.885	−0.059	0.953
	>376	7 (17)	2.437 (0.500–11.888)	0.809	1.102	0.271
Area of tree cover in the SA1	No	12 (63)				
	Yes	6 (9)	6.450 (1.236–33.656)	−0.843	2.212	0.027 ^1^
Land for a commercial facility in the SA1	No	8 (17)				
	Yes	10 (55)	0.445 (0.126–1.569)	0.643	−1.259	0.208
Land for a recreational and cultural facility in the SA1	No	12 (31)				
	Yes	6 (41)	0.246 (0.067–0.911)	0.667	−2.100	0.036 ^1^

^1^ *p*-value is less than 0.05.

**Table 7 vetsci-10-00697-t007:** Univariate conditional logistic models of canine leptospirosis and socioeconomic variables, adjusted by dog population per Statistical Areas (Level 1) (SA1) in 1km buffers around case SA1s (*n* = 18) from December 2017 to January 2023.

Variables		Count of SA1s	Odds Ratio (95% CI)	SE	Z Value	*p* Value
Distance to the nearest veterinary hospital (m)	≤395	9 (58)				
	>395	9 (14)	9.35 (1.617–54.098)	0.895	2.497	0.013 ^1^
The Index of Economic Resources (IER)	≤876	8 (11)				
	>876, ≤931	2 (22)	0.017 (0.001–0.310)	1.487	−2.748	0.006 ^1^
	>931, ≤997	4 (17)	0.088 (0.007–1.045)	1.261	−1.925	0.054
	>997	4 (20)	0.036 (0.001–0.867)	1.626	−2.048	0.041 ^1^
The Index of Education and Occupation (IEO)	≤1141	7 (11)				
	>1141, ≤1179	6 (20)	0.343 (0.052–2.250)	0.960	−1.115	0.265
	>1179, ≤1188	2 (13)	0.087 (0.009–0.894)	1.187	−2.054	0.040 ^1^
	>1188	3 (28)	0.074 (0.010–0.559)	1.034	−2.522	0.012 ^1^

^1^ *p*-value is less than 0.05.

**Table 8 vetsci-10-00697-t008:** Multivariate conditional logistic models of canine leptospirosis and landscape variables, adjusted by dog population per Statistical Areas (Level 1) (SA1) in 1km buffers around case SA1s (*n* = 18) from December 2017 to January 2023. The Akaike Information Criterion (AIC) of the model was 53.484.

Variables		Count of SA1s	Odds Ratio (95% CI)	SE	Z Value	*p* Value
Area of tree cover in the SA1	No	12 (63)				
	Yes	6 (9)	5.797 (1.116–30.106)	0.841	2.091	0.037 ^1^
Land for a recreational and cultural facility in the SA1	No	12 (31)				
	Yes	6 (41)	0.255 (0.064–1.018)	0.707	−1.935	0.053

^1^ *p*-value is less than 0.05.

**Table 9 vetsci-10-00697-t009:** Multivariate conditional logistic models of canine leptospirosis and socioeconomic variables, adjusted by dog population per Statistical Areas (Level 1) (SA1) in 1km buffers around case SA1s (*n* = 18) from December 2017 to January 2023. The Akaike Information Criterion (AIC) of the model was 44.463.

Variables		Count of SA1s	Odds Ratio (95% CI)	SE	Z Value	*p* Value
Distance to the nearest veterinary hospital (m)	≤395	9 (58)				
	>395	9 (14)	4.190 (0.619–28.344)	0.975	1.469	0.142
The Index of Economic Resources (IER)	≤876	8 (11)				
	>876, ≤931	2 (22)	0.024 (0.001–0.585)	1.627	−2.289	0.022 ^1^
	>931, ≤997	4 (17)	0.147 (1.585 × 10^−3^–13.71)	2.313	−0.828	0.140
	>997	4 (20)	0.005 (2.116 × 10^−6^–10.13)	3.924	−1.370	0.052

^1^ *p*-value is less than 0.05.

## Data Availability

The dataset used in the current study will be available upon request.

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
