# Peer review of "Landscape, Socioeconomic, and Meteorological Risk Factors for Canine Leptospirosis in Urban Sydney (2017–2023): A Spatial and Temporal Study"

_vetsci, 2023, doi:10.3390/vetsci10120697_

Round 1

Reviewer 1 Report

Comments and Suggestions for Authors

This manuscript present an interesting investigation into the risk factors associated with canine leptospirosis in Sydney. The author's analysis include various socioeconomic and geographical factors within the studied area. While my limited understanding of statistical methodologies may have hindered a more thorough evaluation of the methodology employed in this study, the information related to Leptospira and leptospirosis was clear and accurate. Furthermore, the authors correctly relate their findings to the existing literature in the discussion, demonstrating a well-informed perspective.

Author Response

Thank you for your review. No comments were made that need to be addressed.

Reviewer 2 Report

Comments and Suggestions for Authors

Title: Landscape and socioeconomic risk factors for canine leptospirosis in urban Sydney (2017-2023)

I would like to commend the authors of this work discussing something important regarding public health and also animal welfare.

Leptospirosis is an important zoonotic pathogen that has several known strains that circulate.

I have a few comments on this manuscript below.

Introduction

Lines 70-71: Previous human studies closely associated leptospirosis with lower socioeconomic status, which reflects suboptimal hygiene and inaccessibility to hospital resources [20, 21].

This is potentially more likely an occupation disease and confounded by social economic status. In Australia, like many other developed world, high social economic people and the poor are likely to use the similar dog parks etc…

Lines 74-76: Following a similar logic, it is reasonable to assume that companion dogs who live in communities with a poorer socioeconomic status could be at higher risk of being infected with Leptospira.

This all depends on where dogs get their infections from.

Methods

Line 163: For conditional logistic regression, the selected case and control SA1s were included and stratified by individual 1km buffers. The clogit() function from survival R package was used for modelling.

What was the ratio of cases to controls? Please explain if this was a matched case control etc. Make it easier to understand all the bots and nuts around the methods.

If I read your results correctly, it appears that you had 18 cases of leptospirosis matched to 72 controls which is 1:4. I would expect this information in the methods.

Lines 30-31: SA1, stratified by 1km 30 buffers surrounding cases (18 cases, 82 controls); General Additive Models of Normalized Differ.

Lines 178-180: From the included residential SA1s (n = 906), 72 control SA1s were randomly selected from 1km 179 buffers around the 18 case SA1s for conditional logistic regression analysis (Figure 1b).

Why are the numbers of controls in lines 30-31 different from those in lines 178-180? Are the numbers representing different models etc?

Lines 154-155: For Poisson regression, all residential SA1s were included for analysis and glm() function from stats Rpackage was used for modelling. Log transformed SA1 dog population was added as an off-set to every Poisson model produced.

I understand that this was a multi-year study and also with certain veterinary clinics involved. How did you account for clustering of cases within a veterinary clinic and also potential repetitive collection of samples on the same clinic?

I am was of the view that a mixed effect Poisson regression model was more suited to analyse these data. Mixed effects models can account for clustering etc…

The same can be said about your case control study. You didn’t account for clustering either. Are there special circumstances why didn’t you adjust for the clustering. Please help others to replicate your methods.

Lines 131-132: Veterinary Practitioners Board of New South Wales website was accessed to retrieve addresses of veterinary hospitals which were then manually geocoded [35].

How many veterinary hospitals were involved in this study?

Results

Why did you put figures and tables right next to the texts? This is making a lot difficult to read.

As I said before above, leptospirosis is an infectious pathogen that is well known and described in literature, I am surprised why your study does not mention what strains were found in your study.

The use of sero-prevalence data alone can be very misleading if you are dealing with antibody titres that doesn’t represent an acute infection e.g. titres less than 1:1600 etc. My understanding is that titres titers of >1:1600 to vaccinal or nonvaccinal serovars can persist 1 year after vaccination. Therefore, although a single positive MAT result may be suggestive of leptospirosis, convalescent-phase testing is still recommended. I am not sure how you handled this in your analysis. I was expecting the descriptive stats that show what proportion of your cases were recent infections versus old infections etc. That way it would show some temporal and spatial trends. This classification would be more informative to public health and also the veterinary industry.

Discussion

Line 260: We found that canine leptospirosis was strongly related to higher seroprevalence in some neighbour hoods, which could indicate higher exposure to Leptospira spp. resulting in clinical leptospirosis.

Did you have any baseline data for dog vaccination in both populations (cases/controls). That would be important because there’s a commercially available vaccine for canine leptospirosis. This alone could help the veterinary industry together with the council to put in mechanisms that would prevent infections. Also, what strains were mostly associated with sero-prevalence? In your methods, you chose to use a titre level cut off of (MAT) titre 1/800 to be sero-positive for the control dogs? Is that the standard recommended cut off for non-vaccinated canine? My understanding that a 4-fold rise in titer, or a seroconversion to ≥1600 is an indication of a current leptospirosis infection. Given that most of the dogs in 2021 during Covid-19 lockdowns, it is possible that dogs could have been bought from very far away breeders already seropositive. Obviously not all of them but a proportion. It is possible that some of the cases were likely to be a false case. I understand that most of the cases/counts of leptospirosis were diagnosed by many systems including PCR. How many strains of leptospirosis were circulating in this study population?

I understand that this study was conducted in Sydney which should mostly urban. Did you consider livestock and vermin as the most likely sources of canine leptospirosis’? The triangle between humans, wildlife/domestic animals and the environment are the three most important aspects to the infections/tansmission of leptospirosis. Sometimes the direction is straightforward but on other times it is a very complicated relationship. To make it simple, I would be interested if you could try to re-analyse the dataset to see if you could separate recent infections from old cases etcs. That way it will be easier to explain some of your findings.

I am not sure if your study has demonstrated the actual risk factors associated with leptospirosis infections in the canine populations.

There are likely more important sources of infections that are needed to be investigated because they’re likely to be of public health importance and relevance. You could have sourced data from the councils about rats/mice infestations etc…and overlaying it on your case/controls etc. That would have provided useful information.

I suggest that authors reanalyse the data with suggested methods to improve the findings and reproducibility.

Author Response

Please find attached a response to reviewer's comments.

Reviewer 3 Report

Comments and Suggestions for Authors

THE GOAL OF THIS STUDY IS TO HIGHLIGHT URBAN RISK FACTORS (LANDSCAPE, SOCIOECONOMIC STATUS AND SEROPREVALENCE) AND CANINE  LEPTOSPIROSIS.

INTERESTING BUT VACCINE STATUS OF CANINE POPULATION WAS NOT TAKEN INTO ACCOUNT AND WOULD BE AFFORD A BIAS.

SEROPREVALENCE WAS ESTIMATED BY L 134 by generating 1km buffers around positive canine samples from sero-survey  performed in 2019 ; BUT VACCINE STATUS OF THE HEALTHY SAMPLED DOGS IS MISSING. HEALTH DOES NOT MEAN THAT THE DOGS ARE NOT INFECTED. VACCINE PREVENTS AGAINST ACUTE AND SEVERE CLINICAL CASES BUT NOT AGAINST INFECTION. MOREOVER INFECTION IN VACCINATED DOGS BOOSTS THE PREVIOUS VACCINE INDUCED ANTIBODIES .  

THEREFORE,  LACK OF THIS DATA COULD INTRODUCE A CONFUSING FACTOR. L 319 a similar association also exists between owner’s education level and attitude towards veterinary prevention against canine leptospirosis. EDUCATION LEVEL IS OFTEN ASSOCIATED WITH BETTER PREVENTIVE CARES BY THE OWNERS AND THEREFORE A HIGHER SEROPREVALENCE IN THESE ANIMALS.

WHEN VACCINE STATUS IS DECREASING IN A POPULATION , ONSET OF PSEUDO OUTBREAK IS FREQUENT.  L77 In December 2017, canine leptospirosis cases re-emerged in Sydney after approximately forty years of apparent absence of disease and the outbreak has been ongoing ever since [22, 23].

MOREOVER L 269  possible underreporting in the two council areas,. THE UNDERREPORTING IS NOT  POSSIBLE. AS EVERYWHERE IN THE WORLD, CANINE LEPTOSPIROSIS IS NOT WELL KNOWN AS CLINICAL CASES ARE GENERALLY MILD CASES WHEN VACCINATED DOGS ARE INFECTED. VET SURGEONS USE WIDELY BACTERINS AND THEREFORE MISSDIAGNOSE CLINICAL CASES OF VACCINATED ANIMALS.

DISCUSSION WOULD BE IMPROVED

2.14.0.0

Author Response

(The authors gave the same response as above.)

Reviewer 4 Report

Comments and Suggestions for Authors

The approach of using landscape and socioeconomic data resources to better understand disease risk is very useful, but in this study the cohort of cases is too small and the analysis uses poorly designed parameters (e.g. using distance to bat colony fails to account for foraging patterns that can extend 10s of km from the colony; distance to veterinary care is over relatively short distances; and there is also failure to contextualise a 'low' socioeconomic in inner Sydney and making perhaps unreasonable comparisons to 'low' socioeconomic groups elsewhere).

The analysis does not add to existing published information where serological data implicates Leptospira interrogans serovar Copenhageni, with rats being the likely reservoir host and it also probably excludes bats.

The combination of previous published information on this outbreak and the lack of new insights provided by this analysis means that I am reluctant to recommend this manuscript for publication.

Comments on the Quality of English Language

The paper is well written and understandable.

line 46: "Gram" not "gram"

Author Response

(The authors gave the same response as above.)

Reviewer 5 Report

Comments and Suggestions for Authors

The article it is in condition to be published

Author Response

The reviewer is thanked for their recommendation to publish.

Round 2

Reviewer 2 Report

Comments and Suggestions for Authors

Thanks for addressing the issues that were raised. .

All the best of luck.